Exploration of immunoglobulin transcriptomes from mice immunized with three-finger toxins and phospholipases A2 from the Central American coral snake, Micrurus nigrocinctus

Laustsen Andreas H. ahola@bio.dtu.dk 1
Engmark Mikael 1 2
Clouser Christopher 3 4
Timberlake Sonia 5
Vigneault Francois 3 4
Gutiérrez José María 6
Lomonte Bruno 6
1 Department of Biotechnology and Biomedicine, Technical University of Denmark , Kgs. Lyngby , Denmark
2 Department of Bio and Health Informatics, Technical University of Denmark , Kgs. Lyngby , Denmark
3 Juno Therapeutics , Seattle , WA , United States of America
4 AbVitro , Boston , MA , United States of America
5 Finch Therapeutics , Somerville , MA , United States of America
6 Instituto Clodomiro Picado, Universidad de Costa Rica , San José , Costa Rica
Sanderson J. Thomas
Electronic publication date: 2017 Jan 24
Publication date: 2017
Volume: 5
Electronic Location ID: e2924
Received 2016 Aug 15; Accepted 2016 Dec 19
Copyright: ©2017 Laustsen et al.
Copyright year: 2017
Copyright holder: Laustsen et al.
License: This is an open access article distributed under the terms of the Creative Commons Attribution License, which permits unrestricted use, distribution, reproduction and adaptation in any medium and for any purpose provided that it is properly attributed. For attribution, the original author(s), title, publication source (PeerJ) and either DOI or URL of the article must be cited.
License URL: https://creativecommons.org/licenses/by/4.0/

Keywords: Murine immune response, Venom, Micrurus nigrocinctus, High-throughput sequencing, Antivenom, Antibody isotyping, Toxins, Coral snake, Antibodies, Transcriptomics

Funding: Juno Therapeutics Inc., Instituto Clodomiro Picado, Universidad de Costa Rica Novo Nordisk Foundation NNF13OC0005613 NNF16OC0019248 The following institutions and foundations supported the research: Juno Therapeutics Inc., Instituto Clodomiro Picado, Universidad de Costa Rica, and the Novo Nordisk Foundation (NNF13OC0005613 and NNF16OC0019248). The funders had no role in study design, data collection and analysis, decision to publish, or preparation of the manuscript.

==============================
Snakebite envenomings represent a neglected public health issue in many parts of the rural tropical world. Animal-derived antivenoms have existed for more than a hundred years and are effective in neutralizing snake venom toxins when timely administered. However, the low immunogenicity of many small but potent snake venom toxins represents a challenge for obtaining a balanced immune response against the medically relevant components of the venom. Here, we employ high-throughput sequencing of the immunoglobulin (Ig) transcriptome of mice immunized with a three-finger toxin and a phospholipase A2 from the venom of the Central American coral snake, Micrurus nigrocinctus. Although exploratory in nature, our indicate results showed that only low frequencies of mRNA encoding IgG isotypes, the most relevant isotype for therapeutic purposes, were present in splenocytes of five mice immunized with 6 doses of the two types of toxins over 90 days. Furthermore, analysis of Ig heavy chain transcripts showed that no particular combination of variable (V) and joining (J) gene segments had been selected in the immunization process, as would be expected after a strong humoral immune response to a single antigen. Combined with the titration of toxin-specific antibodies in the sera of immunized mice, these data support the low immunogenicity of three-finger toxins and phospholipases A2found in M. nigrocinctusvenoms, and highlight the need for future studies analyzing the complexity of antibody responses to toxins at the molecular level.

Introduction

Snakebite envenomings represent a major public health concern in tropical regions of the world (Williams et al., 2011). Despite emerging discoveries that may one day pave the way for novel biotechnology-based antivenoms (reviewed by Laustsen et al., 2016a; Laustsen et al., submitted), animal serum-derived antivenoms remain the cornerstone of snakebite envenoming treatment (Gutiérrez et al., 2011). Production of antivenom is challenged by a large variation in immunogenicity of many key snake venom toxins resulting in unpredictable immune responses in production animals (Cook et al., 2010; Guidolin et al., 2010). It has been shown that many of the immunogenic venom components are in fact not important for toxicity (Antúnez et al., 2010; Gutiérrez et al., 2009; Laustsen et al., 2015), and conversely, that some highly toxic venom components, such as α-neurotoxins (both short and long neurotoxins), phospholipases A2, and P-I snake venom metalloproteinases may be poorly immunogenic (Schottler, 1951; Gutiérrez et al., 2009; Chotwiwatthanakun et al., 2001; Ownby & Colberg, 1990; Judge et al., 2006; Wong, Tan & Tan, 2016; Tan et al., 2016; Leong et al., 2015; Tan et al., 2015). Combined, this creates a challenge for antivenom production, since the goal of obtaining an antivenom with a strong, yet balanced response against all the medically relevant toxins becomes a complex endeavor.

Coral snakes (genera Micrurus, Leptomicrurus, and Micruroides) are, together with the sea snake Hydrophis (Pelamis) platura, the representatives of the snake family Elapidae in the Americas, comprising approximately 85 species (Campbell & Lamar, 2004; the Reptile Database—www.reptile-database.org). Although Micrurus species are only responsible for about 1-2% of snakebite cases in this continent, roughly corresponding to 750 to 1000 cases per year, envenomings by these snakes can be fatal if not treated properly and timely (Warrell, 2004; Gutiérrez et al., 2016; Bucaretchi et al., 2016). Envenomings resulting from coral snakebites are predominantly associated with descending neuromuscular paralysis, which may end in respiratory arrest (Warrell, 2004; Bucaretchi et al., 2016).

Production of antivenoms against Micrurus snakes is particularly challenging, as (a) it is very difficult to maintain coral snakes in captivity (Chacón et al., 2012); (b) the majority of Micrurus species provide a very low yield of venom, implying that the collection of the quantities of venom required for horse immunization and quality control testing demands the ‘milking’ of many specimens (Chacón et al., 2012; Bolaños, 1972); and (c) there is a variable extent of immunological cross-recognition between venoms from coral snakes of different species; hence, antivenoms raised against some species are not always effective in the neutralization of venoms of other species (Bolaños, Cerdas & Abalos, 1978; Tanaka et al., 2016). As a result, only a few laboratories manufacture Micrurus antivenoms, and several countries where these snakes inhabit completely lack this therapeutic resource, e.g., Venezuela, Ecuador, Peru, Bolivia, the Guyanas, and Paraguay, which severely limits the clinical management of these accidents.

Knowledge on the composition of the venoms of Micrurus species has increased steadily over the last years, as a consequence of proteomic characterizations (reviewed by Lomonte et al., 2016b). Two main venom phenotype patterns have been identified, i.e. venoms rich in neurotoxins of the three-finger toxin (3FTx) family, and venoms rich in phospholipases A2 (PLA2s) (Fernández et al., 2015). In addition to these two main protein families, other minor components of these venoms include L-amino acid oxidases, serine proteinases, metalloproteinases, nerve growth factor, C-type lectin-like proteins, Kunitz-type inhibitors, among others (Fernández et al., 2011; Fernández et al., 2015; Corrêa-Netto et al., 2011; Lomonte et al., 2016a; Sanz et al., 2016; Rey-Suárez et al., 2011; Rey-Suárez et al., 2016). In some cases, the toxins playing the main role in overall toxicity have been identified, these being 3FTxs and PLA2s (Rey-Suárez et al., 2012; Vergara et al., 2014; Fernández et al., 2015; Castro et al., 2015; Ramos et al., 2016).

The limited immunogenicity of the highly toxic PLA2s and 3FTxs (Fernández et al., 2011; Rosso et al., 1996; Alape-Girón et al., 1996) represents another difficulty in production of Micrurus antivenom, since it thwarts the goal of raising a balanced immune response against these medically relevant toxins. In order to further explore how these toxins interact with the mammalian immune system, we chose a mouse model and employed an NGS approach using the AbSeq™ technology developed by AbVitro (now Juno Therapeutics, https://www.junotherapeutics.com), based on Illumina sequencing (Fig. 1). The methodology was utilized to sequence immunoglobulin (Ig) encoding mRNA transcripts from splenic B-lymphocytes in mice subjected to immunization with either a 3FTx or a PLA2 toxin from the venom of M. nigrocinctus (Central American coral snake). By this approach, the transcription levels of different immunoglobulin isotypes and dominant clones of B-lymphocytes with a particular usage of V (variable) and J (joining) gene segments can be determined for Ig heavy chain transcripts. This methodology has previously been employed for investigating B-cell populations in autoimmune (Stern et al., 2014) or infectious diseases (Tsioris et al., 2015; Di Niro et al., 2015), for example. By employing the AbSeq™ high-throughput approach, we explore, for the first time, the Ig transcriptome including VJ usage patterns in individual animals subjected to immunization with two relevant toxin classes of elapid snakes. Although exploratory in its nature and somewhat limited by a small sample size, this study thus provides novel insight into the humoral response of mice immunized with 3FTx or PLA2 toxins and highlights important challenges of raising antibodies against poorly immunogenic toxins.

Figure 1 Schematic overview of the experimental strategy.

Following extraction from the snakes, venom is fractionated by HPLC, and the fractions of interest are used for immunization of rodents. Upon completion of the immunization protocol, the rodents are sacrificed and RNA is extracted and subjected to the AbSeq protocol (see ‘Materials and Methods’), whereby the RNA is reverse transcribed and barcoded, allowing for correct pairing of VH and VL chains after DNA sequencing.

Materials and Methods

Snake venom and toxins

Venom from M. nigrocinctus was obtained from a pool of more than 50 adult specimens collected in the Central Pacific region of Costa Rica, kept at the serpentarium of Instituto Clodomiro Picado, Universidad de Costa Rica. The snakes were not collected for this study, but belong to Instituto Clodomiro Picado, where their venoms are routinely used for the production of antivenom, wherefore a field permit was not necessary. Venom extraction is performed every four months (Chacón et al., 2012). The venom was lyophilized and stored at −20 °C.

Fractionation of the venom was performed by RP-HPLC on a C18 column (4.6 × 250 mm, 5 µm particle diameter; Supelco) as previously described (Fernández et al., 2011). In brief, 2 mg of venom dissolved in 200 µL of water containing 0.1% trifluoroacetic acid (TFA; solution A) were separated at 1 mL/min in an Agilent 1200 chromatograph monitored at 215 nm, applying a gradient towards solution B (acetonitrile, containing 0.1% TFA): 0% B for 5 min, 0–15% B over 10 min, 15–45% B over 60 min, 45–70% B over 10 min, and 70% B over 9 min. Fractions of interest (the major components of the venom, belonging to the 3FTx (lethal) and PLA2 (myotoxic) protein families) were collected manually, dried in a vacuum centrifuge, and identified by trypsin digestion followed by MALDI-TOF/TOF mass spectrometry (Fernández et al., 2011). Proteins were redissolved in water and their concentrations were estimated on the basis of their absorbance at 280 nm, using a NanoDrop (Thermo) instrument.

Immunization of mice

Three female CD-1 mice (16–18 g) from Instituto Clodomiro Picado were immunized with a three-finger toxin (3FTx), and two with a phospholipase A2 (PLA2), respectively. These correspond to fractions #3 (∼P80548) and #30 (∼P81166/P81167) described in the previous proteomic characterization of this venom (Fernández et al., 2011). All toxin doses were injected by the intraperitoneal route. The priming dose was 1 µg emulsified in Freund’s complete adjuvant, followed by five booster doses injected in physiological saline without adjuvant, at days 15 (1 µg), 43 (2 µg) 63 (4 µg), and 83 (6 µg for the 3FTx and 8 µg for the PLA2). At day 90, after obtaining a blood sample for monitoring of the antibody response by enzyme-immunoassay, mice were euthanized by CO2 inhalation. Their spleens were immediately removed, cut in small pieces, and disaggregated over a stainless steel mesh to obtain splenocytes. These cell suspensions were aliquoted in RNAlater® solution (Thermo) and shipped within 24 h to AbVitro, at room temperature, for subsequent molecular studies. The use of animals for these experiments followed the ethical guidelines of the Comité Institucional para el Uso y Cuido de Animales (CICUA), Universidad de Costa Rica, with the approval number 82-08.

Enzyme-immunoassay (ELISA)

In order to evaluate the individual antibody responses of the mice, wells in MaxiSorp 96-well plates (NUNC, Roskilde, Denmark) were coated overnight with 1 µg of either 3FTx or PLA2, dissolved in 100 µL PBS (0.12 M NaCl, 0.04 M sodium phosphate, pH 7.2). Wells were washed five times with PBS and blocked by adding 100 µL PBS containing 2% (w:v) bovine serum albumin (BSA; Sigma), and incubated at room temperature for 1 h. Plates were then washed five times with PBS. Serial dilutions of serum from each mouse were prepared in PBS + 2% BSA and 100 µL was added to each well, in triplicates, and incubated overnight at 4 °C. Normal mouse serum, run simultaneously under identical conditions was used as a control for background. Plates were then washed five times with PBS, followed by the addition of 100 µL of a 1:3,000 dilution of anti-mouse IgG (whole molecule) antibodies conjugated to alkaline phosphatase, in PBS + 1% BSA. The plates were incubated for 2 h, and then washed five times with FALC buffer (0.05 M Tris, 0.15 M NaCl, 20 µM ZnCl2, 1 mM MgCl2, pH 7.4). Development of color was attained by addition of 100 µL p-nitrophenyl phosphate (1 mg/mL in 9.7% v/v diethanolamine buffer, pH 9.8) and absorbances at 405 nm were recorded (Multiskan FC; Thermo Scientific).

Assessment of mRNA quality

Assessment of RNA quality was performed using Agilent’s TapeStation according to the manufacturers protocol and algorithm to calculate RINe scores (http://www.agilent.com/cs/library/technicaloverviews/public/5990-9613EN.pdf).

Library preparation and high-throughput sequencing of B-cell receptors

The method for high-throughput sequencing of the B-cell repertoire was performed as described elsewhere (Di Niro et al., 2015; Tsioris et al., 2015). Briefly, RNA was reverse-transcribed into cDNA using a biotinylated oligo dT primer. An adaptor sequence was added to the 3′ end of all cDNA, which contains the Illumina P7 universal priming site and a 17-nucleotide unique molecular identifier (UMI). Products were purified using streptavidin-coated magnetic beads followed by a primary PCR reaction using a pool of primers targeting the IGHA, IGHD, IGHE, IGHG, IGHM, IGKC and IGLC regions, as well as a sample-indexed Illumina P7C7 primer. The immunoglobulin-specific primers contained tails corresponding to the Illumina P5 sequence. PCR products were then purified using AMPure XP beads. A secondary PCR was then performed to add the Illumina C5 clustering sequence to the end of the molecule containing the constant region. The number of secondary PCR cycles was tailored to each sample to avoid entering plateau phase, as judged by a prior quantitative PCR analysis. The final products (reverse-transcribed UTR + VDJ + partial—Cexon segments of the transcripts of the immunoglobulin chains, plus molecular barcode, Illumina multiplexing barcode, and Illumina sequencing adapters) were purified, quantified with Agilent TapeStation and pooled in equimolar proportions, followed by high-throughput paired-end sequencing on the Illumina MiSeq platform. For sequencing, the Illumina 600 cycle kit was used with the modifications that 325 cycles was used for read 1, 6 cycles for the index reads, 300 cycles for read 2 and a 10% PhiX spike-in to increase sequence diversity.

VJ repertoire sequencing data analysis

MiSeq reads were demultiplexed using Illumina software, and processed with the pRESTO toolsuite (Vander Heiden et al., 2014) as following: Positions with less than Phred quality 5 were masked with Ns. Isotype-specific primers and molecular barcodes (UIDs or UMIs) were identified in the amplicon and trimmed, using pRESTO MaskPrimers-cut. A read 1 and read 2 consensus sequence was generated separately for each mRNA from reads grouped by unique molecular identifier, which are PCR replicates arising from the same original mRNA molecule of origin. UMI read groups were aligned with MUSCLE (Edgar, 2004), and pRESTO was used to BuildConsensus, requiring ≥60% of called PCR primer sequences agree for the read group, maximum nucleotide diversity of 0.1, using majority rule on indel positions, and masking alignment columns with low posterior (consensus) quality. Paired end consensus sequences were then stitched in two rounds. First, ungapped alignment of each read pair’s consensus sequence termini was optimized using a Z-score approximation and scored with a binomial p-value as implemented in pRESTO AssemblePairs-align. For read pairs failing to stitch this way, stitching was attempted using the human BCR germline V exons to scaffold each read prior to stitching or gapped read-joining, using pRESTO’s AssemblePairs-reference. Positions with posterior consensus quality less than Phred 5 were masked again with Ns. Each mRNA was annotated for V, D, J germline gene of origin, productivity, and CDR3 region using igblastn (Ye et al., 2013). Igblast output was parsed and analyzed with custom scripts as described below (proprietary to AbVitro (now Juno Therapeutics)) and visualized with R (R Core Team, 2014). Briefly, clones were defined using a conservative approach, grouping mRNAs from the same V and J germline gene of origin and having the same isotype and CDR3 nucleotide sequence. Non-productive rearrangements (as predicted by Igblast) were excluded from the analysis. Within each sample, clones were ranked by abundance (mRNAs/clone) and plotted to highlight the most expanded binding motifs. To examine preferences for V–J usage common across mice exposed to the same toxin, clones were grouped into larger bins that each encompass a single V gene-J gene combination, but all isotypes and CDR3s.

In silico epitope predictions

The sequences of selected toxin components in the two venom fractions used for immunization were obtained from the UniProtKB database (http://www.uniprot.org) and linear parts of B-cell epitopes were predicted using the Bepi-Pred 1.0 server (Larsen, Lund & Nielsen, 2006) using 0.9 as Threshold to obtain a sensitivity of 0.25 and Specificity of 0.91. As no experimental structures were available, homology models were build based CPHmodels 3.2 (Nielsen et al., 2010) and the pdb-files were submitted to DiscoTope 2.0 (Kringelum et al., 2012) using 0.5 as Threshold to obtain a sensitivity of 0.23 and Specificity of 0.90.

Results and Discussion

Three-finger toxins (3FTx) and phospholipase A2s (PLA2) are the two most abundant toxin families in the venom of M. nigrocinctus (Fernández et al., 2011), and generally they are the two snake toxin families which have been most investigated (Laustsen et al., 2016a). In the venom of M. nigrocinctus these toxins cause neuromuscular paralysis, owing to a combination of pre- and post-synaptic actions, and myotoxicity, providing the venom with its high toxicity (Rosso et al., 1996; Alape-Girón et al., 1996). In previous studies it was observed that 3FTxs and PLA2s were recognized more weakly than larger proteins from this venom, by a therapeutic equine antivenom (Fernández et al., 2011). Despite their low immunogenicity, it was possible to raise an antibody response against both toxins in four out of five mice, although high variation in the antibody titer was observed (Fig. 2). Mice immunized with PLA2 had a higher antibody titer than mice immunized with the 3FTx, in agreement with the higher molecular mass of the former.

Figure 2 ELISA titrations of serum antibodies against M. nigrocinctus PLA2 or 3FTx in mice following a 90-day immunization protocol.

Two mice were immunized with PLA2, and three were immunized with 3FTx. Plates were coated with either PLA2 or 3FTx, and antibodies were detected as described in ‘Materials and Methods’.

Assessment of the mRNA from harvested mouse splenocytes indicated that it was of sufficient quality to proceed to sequencing (RINe scores between 5.2 and 6.4). A high-throughput sequencing approach (AbSeq™) was employed to investigate transcription levels of Ig isotypes and the usage of V and J gene segments for heavy chain assembly in mice that were immunized with a 3FTx or a PLA2. Investigation of the 50 most frequent VJ combinations for the immunized mice did not, however, result in identification of a dominant combination, as the VJ usage was found to be similar across all samples (Fig. 3). This finding suggests that the generated antibody responses might be diverse and that multiple specific antibodies with low abundance are generated in each mouse.

Figure 3 Comparison of the relative abundance of mRNA for the 50 most abundant VJ combinations for the mouse 3FTx-3 and mouse PLA2-2 showing VJ usage to be similar across samples.

Similar VJ usage patterns were observed for other pairs of immunized mice.

Looking at the sequences of all mRNA transcripts encoding heavy chain variable domain (VH) clones across each sample, we were able to find shared VH clones with similar relative abundances in either the PLA2-immunized or the 3FTx-immunized mice (Fig. 4). In comparison, almost no VH clones were shared between mice immunized with different toxins (Fig. 5). This implies that the immunization procedure did indeed elicit specific, but different responses dependent on whether PLA2s or 3FTxs were employed for immunization. The relatively high number of VH clones found in both of the PLA2-immunized mice (Fig. 4A) compared to lower number of VH clones found across the three 3FTx-immunized mice (Figs. 4B–4D) further indicate that immunization with PLA2s is more prone to give rise to antibodies transcribed in similar quantities. Also, an intermediate number of similar VH clones was found in both the PLA2-1 and 3FTx-3 samples (Fig. 5E), even though the correlation in relative abundance was not equally pronounced. This is likely explained by the fact that the majority of VH clones found in both PLA2-immunized mice are not expected to be specific towards the toxins, but instead are likely to be directed against other (environmental) antigens that the mice have encountered throughout their life.

Figure 4 Relative abundance of unique VH clone transcripts compared between samples.

A large group of VH transcripts are found in similar abundance in different mice immunized with the same toxin. (A) Comparison between mouse PLA2-1 and PLA2-2, (B) Comparison between mouse 3FTx-1 and 3FTx-2, (C) Comparison between mouse 3FTx-1 and 3FTx-3, (D) Comparison between mouse 3FTx-2 and 3FTx-3.

Figure 5 Relative abundance of unique VH clone transcripts compared between samples.

Only few VH transcripts are found in similar abundance in more than one mouse, when mice immunized with different toxins are compared. (A) Comparison between mouse PLA2-1 and 3FTx-1, (B) Comparison between mouse PLA2-2 and 3FTx-1, (C) Comparison between mouse PLA2-1 and 3FTx-2, (D) Comparison between mouse PLA2-2 and 3FTx-2, (E) Comparison between mouse PLA2-1 and 3FTx-3, (F) Comparison between mouse PLA2-2 and 3FTx-3.

Figure 6 Overview of total mRNA transcripts encoding different immunoglobulin isotypes from the immunized mice (normalized).

Numbers above each bar represents the mRNA count in each sample.

The AbSeq™ antibody sequencing methodology is capable of determining the Ig isotype of the identified VH clones. The coloring of the VH clones in Figs. 4 and 5 reveals that a large number of the most abundant VH clones present in the mice are of the IgA isotype, which was confirmed by further investigation of all mRNA transcripts from the splenocytes (Fig. 6). This is surprising, as IgG is known to be the dominant immunoglobulin class in mouse blood after the response to T-dependent protein antigens. The observation could possibly be explained by differences in expression levels due to different translation rates and half-lives of mRNA transcripts encoding different immunoglobulin isotypes. All approved antibody-based therapies on the market are based on IgGs (Walsh, 2014), which are also the desired isotype for antivenoms. In immunized horses for antivenom production, two isotypes of IgG are largely responsible for the neutralization of toxic effects in the case of viperid snake venoms (Fernandes et al., 2000). However, analysis of the transcripts obtained from the immunized mice revealed only a low percentage of IgG transcripts, as compared to the transcripts for other Igs (Figs. 4–6). With the acknowledgement of the small sample size employed in this study, this finding may indicate a difficulty in raising potent IgG antibodies against both 3FTxs and PLA2s in rodents. If similar difficulty is present in horses, this may therefore have implications for antivenom production. Our results may further suggest that the immune response is slightly lower against 3FTxs than for PLA2s based on the lower abundance of IgG transcripts in mice immunized with 3FTx (Fig. 6). Taken together with the results from the ELISA assay (Fig. 2) and the observation that immunization with PLA2s is more prone to give rise to similar Ig transcripts (Fig. 4A vs. Figs. 4B–4D), we suggest that the PLA2 toxins may possibly be slightly more immunogenic than the 3FTx, although neither toxin seems to have high immunogenicity. The underlying reason for this could possibly be due to the smaller molecular size of 3FTx compared to PLA2s, or that PLA2s may contain distinct epitopes better capable of eliciting an adaptive immune response than 3FTxs. Predictions of possible B-cell epitopes using the protein sequences and BepiPred 1.0 (Larsen, Lund & Nielsen, 2006) or DiscoTope 2.0 (Kringelum et al., 2012) with structural models of the investigated toxins as input do not indicate a major difference in the bare number of possible epitopes. However, the suggested difference in immunogenicity is further indicated by the fact that only two IgG-encoding mRNA transcripts are found in the top 20 most abundant Ig-encoding mRNA transcripts for only one out of three of the 3FTx-immunized mice. In comparison, six and nine of the top 20 mRNA transcripts for mice immunized with PLA2s encode the IgG isotype (Fig. 7). It would be interesting to assess whether the immune response of horses against these elapid venom toxins is also characterized by a low proportion of IgG—a finding that would have evident implications for antivenom manufacture, as IgG has been shown to be the antibody isotype of therapeutic value (Fernandes et al., 2000). However, this is beyond the scope of this exploratory study.

Figure 7 The 20 most abundant VH clone transcripts and their corresponding isotypes in each immunized mouse based on their fraction of total immunoglobulin mRNA.

Concluding Remarks and Outlook

In addition to demonstrating the utility of high-throughput sequencing technology, AbSeq™, for investigation of immune responses in animals immunized with snake venom toxins, the exploratory findings presented here may indicate possible difficulties in obtaining an IgG response against the medically important toxins of the 3FTx and PLA2 families from M. nigrocinctus. Given that these proteins play key toxic roles in envenomings by elapid snakes, this underlines a drawback of current antivenom production based on immunized animal serum, since IgG has been shown to be the antibody isotype of therapeutic value (Fernandes et al., 2000). Although based on a small exploratory sample size, these findings therefore contribute to the understanding of snake toxin immunogenicity and indicate the possible difficulty in obtaining balanced immune responses in animals during the immunization process.

We thank Mikael Rørdam Andersen from the Technical University of Denmark for fruitful scientific discussion.

Additional Information and Declarations

Competing Interests

Author Contributions

Animal Ethics

Data Availability

Bruno Lomonte is an Academic Editor for PeerJ. Christopher Clouser and Francois Vigneault are employees of Juno Therapeutics, Seattle, Washington, United States of America, and AbVitro, Boston, United States of America. Sonia Timberlake is an employee of Finch Therapeutics, Somerville, Massachusetts, United States of America. The authors declare no other competing interests.

Andreas H. Laustsen conceived and designed the experiments, analyzed the data, wrote the paper, prepared figures and/or tables, reviewed drafts of the paper.

Mikael Engmark analyzed the data, wrote the paper, prepared figures and/or tables, reviewed drafts of the paper.

Christopher Clouser and Sonia Timberlake performed the experiments, analyzed the data, contributed reagents/materials/analysis tools, prepared figures and/or tables, reviewed drafts of the paper.

Francois Vigneault and José María Gutiérrez conceived and designed the experiments, analyzed the data, contributed reagents/materials/analysis tools, reviewed drafts of the paper.

Bruno Lomonte conceived and designed the experiments, performed the experiments, analyzed the data, contributed reagents/materials/analysis tools, prepared figures and/or tables, reviewed drafts of the paper.

The following information was supplied relating to ethical approvals (i.e., approving body and any reference numbers):

Comité Institucional para el Uso y Cuido de Animales (CICUA), Universidad de Costa Rica.

Approval number 82-08 (experiments on mouse toxicity tests and neutralization by antivenoms).

The following information was supplied regarding data availability:

The raw sequencing data is irrelevant for the reader as this is only used to isotype immunoglobulins—not to deduct any information about specific sequences.

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
