# Peer review of "Exploration of immunoglobulin transcriptomes from mice immunized with three-finger toxins and phospholipases A2 from the Central American coral snake, Micrurus nigrocinctus"

_PeerJ, doi:10.7717/peerj.2924_

## Round 0.1 · original submission · Major Revisions

Dear Authors,

I have carefully read the comments of the reviewers and agree with reviewers 1 and 2 that there are important concerns about the samll sample size and lack of certain positive and negative controls in the mouse study.

If these can be adequately addressed please resubmit a revised version of your manuscript as per instructions on the PeerJ website.

Sincerely

Thomas Sanderson
PeerJ editor

Reviewer 1 ·

Basic reporting

The structure of the manuscript is fine, and the standard of written English is high. References to "next generation sequencing technology" with respect to Illumina platforms is a little old fashioned, as newer generations of technology are now available, and these instances should be modified to "high throughput sequencing technology".

Some minor issues which would need to be addressed:

Introduction
Lines 66-67, provide numbers of snakebite cases in addition to percentage
Lines 71-74, provide references for claims (a) and (b)
Line 79, which countires lack antivenom? How badly are these affected by snakebites form this species?
Lines 89-91, provide examples of toxins that have been identified
Line 96, justify choice of mouse model. This is essential considering most anitvenoms are raised in other, typically large, mammals
Line 97, provide overview of the AbSeq technology/general methodology, ideally as a figure

Methods
Lines 114-115, details of ages and sex of snakes needed, even if approximate (adult, sub-adult, juvenile). Also need to provide more information on frequency of milking, for example whether each snake was milked once, or multiple times
Lines 122-123, justify choice or method of selection of “fractions of interest”
Line 129, justify choice of CD1 strain mice, provide source, age, weight and sex data. Justify sample size and provide statistical evidence (e.g. power calculations) to support this
Line 136 – was CO2 administered in a rising concentration?
Line 202, custom scripts should be briefly described and made available in an online repository

Results and discussion
Line 220, M. nigrocinctus should not be italicised here
Line 253, which other antigens?
Lines 275-277, have such variations in half-life and translation rates been observed?
Lines 289-291, why not try to predict antigenicity of the toxins used?
Line 310, should read “the utility of high throughput sequencing technology”

Experimental design

I have some concerns regarding the experimental design. There is no justification for the small sample size (three mice) and adequate controls appear to be lacking. The authors use normal mouse serum as a control in some of their experiments, when a more appropriate control would be mice injected with saline or similar. Because of this, it is not possible to determine to what extent the observed effects are related to the injected toxins, or the process of handling and repeated injection. Similarly, the apparent lack of IgG response could be biological, or an artefact. Without the use of a highly antigenic control (ideally one known to stimulate an IgG response in mice) an equally valid explanation for the observed results is that the assay is incapable of detecting IgG.

In my view, these issues need to be resolved prior to publication.

Validity of the findings

Without the modifications suggested above, I do not see how the results can be properly interpreted.

Additional comments

No Comments

Reviewer 2 ·

Basic reporting

The article is written clearly in English of professional standards. The structure is acceptable; figures provided are relevant to the content of the article.
A few relevant literature should have been referenced and discussed though. Please refer to General/specific comments to Authors in the following section.

Experimental design

The experimental design is relevant and appropriate to the study. A few comments can be found in the following section of "Comments to Authors".

Validity of the findings

Please see "Comments to Authors" in the following section.

Additional comments

General comments:
The manuscript authored by Laustsen et al. reported the use of next generation sequencing technique to study the immunoglobulin transcriptome of mice immunized with a 3FTx and a PLA2 from the venom of Micrurus nigrocinctus, and drew conclusion on the limited immunogenicity of the two selected toxins tested in mice. The work is straightforward and the methodology employed is suitable for the study design. The data is appropriately analysed but the findings have limited impact and novelty in the field, as the fact of low immunogenicity and immunoneutralization of major elapid toxins has previously been demonstrated in several papers on different species, together with suggestions and trials on how immunization protocol for antivenom production could be improved. These studies on in vivo potency would have made relevant references in correlating and supporting the authors’ observation, unfortunately all credits have been neglected. Nevertheless, the attempt to explore animal’s response to toxin immunization through spleen transcriptome by Laustens et al. is commendable. As novelty and impact are not the major assessment criteria, the manuscript on its own still carries academic merits for publication, after the authors have addressed the queries/issues satisfactorily.

Specific comments:
1. Can the authors please explain how n = 2 and n = 3 for each toxin (immunization) can be statistically valid for their conclusion?

2. On the comparative analysis: It would have been more informative if a toxin of a known good immunogenicity, such as a high molecular weight toxin/enzyme, was included in the study to verify if the low response toward 3FTx is purely because of the small molecular weight. Also, the model of host animal was mice in this study which is known to vary immunologically from the animals commonly used (horses typically) in antivenom production. Can the authors justify or acknowledge these as limitation of their study?

3. Did the authors indicate that the finding is applicable to all 3FTx and PLA2 immunogenicity of all species of elapid lineages? Although speculation is welcome, lacking sufficient variables and number of specimens (mice, and snake species), the data should be interpreted conservatively avoiding generalization. This should be another limitation to state.

4. Pg 6, Ln 58-60: Low immunogenicity and poor neutralization responses to small molecular weight toxins (in particular the short alpha-neurotoxins) and venoms predominated with such toxins, have been shown for several elapid species from Asia. The implication on antivenom production has also been highlighted; moreover, various 3FTx types and PLA2 were covered. These studies provide evidence on in vivo neutralization that should be referenced in this manuscript. Among the few, please see:
a. Wong KY et al. 2016. Venom and purified toxins of the spectacled cobra (Naja naja) from Pakistan: Insights into toxicity and antivenom neutralization. Am J Trop Med Hyg. 94(6):1392-9.
b. Tan KY et al. 2016. Neutralization of the Principal Toxins from the Venoms of Thai Naja kaouthia and Malaysian Hydrophis schistosus: Insights into Toxin-Specific Neutralization by Two Different Antivenoms. Toxins 8 (4), 86.
c. Leong PK et al. 2015. Immunological cross-reactivity and neutralization of the principal toxins of Naja sumatrana and related cobra venoms by a Thai polyvalent antivenom (Neuro Polyvalent Snake Antivenom). Acta tropica 149, 86-93
d. Tan et al. 2015. Venomics of the beaked sea snake, Hydrophis schistosus: A minimalist toxin arsenal and its cross-neutralization by heterologous antivenoms. J Proteomics 126, 121-130

5. Pg 9-10: under the Methodology section: It was not mentioned how and when were the sera of the five mice collected. Please include this in the methods.

6. Pg 10, Ln 150-151: Did the authors collect the sera at different days (post-immunization) or only at the end of the study (over 90 days)? Please specify clearly in the manuscript.

7. Pg 11, Ln 177-179: Are the 'final products' referred to the transcripts of the immunoglobullin chains? What were the sequences and how were these determined of their identity, and quantified? How was the expression level measured (important for comparison between groups)? Please consider providing the sequences and expression levels as supplementary data for referencing use by readers.

8. Results and Discussion. Figure 1. Caption. – specify: serum antibodies of which day?

9. Pg 18, Ln 282, Fig. 5-6: Please explain how the result (high abundance of IgA but low percentage of IgG) can be extrapolated to the current antivenom production where horses are used? (this was mentioned above in General comments and can be answered there)

10. Pg 18: As stated, in my opinion, reports published previously on limited immunogenicity and low potency of antivenoms against elapid toxins from several different species, should be referenced and discussed in this study. Although those studies did not utilize mice spleenocyte transcriptomics, the results were evidenced by ELISA and in vivo neutralization using commercial equine antivenom of IgG derivatives which are used clinically. Credits should be given to these studies, while supporting the Ig NGS work reported in this paper.

11. Conclusion: Pg 20 Ln 312-313. The conclusion based on the finding is appropriate; however, in view of the implication on practice, limitation of the study may be well stated here, that the Results were generated in a mouse model; and involved only one selected 3FTx and one PLA2 from the Central Ameican coral snake, M. nigrocinctus.

Reviewer 3 ·

Basic reporting

This is a well written paper that includes informative, clear figures and the methodology is appropriate to the research objective.

Experimental design

Meets the journal's standards.

Validity of the findings

The findings of this paper are sound. Some assumptions have been made that appear to conflict with the results. Thus, elapid 3FTXs and PLA2s are repeatedly stated as being weakly immunogenic (this is indeed the consensus view) but the Figure 1 data suggests that this is not always the case. The absence of other venom protein immunogens is, I think, a significant oversight to many of the conclusions drawn.

I am also concerned that these findings - from mice immunized with just 2 venom proteins - are used in the concluding paragraph to make wildly sweeping statements about antivenom production, which utilizes mostly horses immunized with whole venom, and often multiple venoms. I suggest that this section be rephrased to avoid unsupported speculation - the results are sufficiently strong that they do not require embellishment.

---

## Round 0.2 · Minor Revisions

Dear authors,

Please see the minor comment of one of the reviewers concerning acknowledging the limitations of the small sample size in the abstract.
With this correction made, I judge the manuscript to be acceptable for publication in PeerJ.

Sincerely
Thomas Sanderson, editor
PeerJ

Reviewer 2 ·

Basic reporting

No comment.

Experimental design

No comment. The experimental design was the same as presented in the previous manuscript.

Validity of the findings

Findings appear to support the known phenomenon of low immunogenicity of snake venom three-finger toxins and phospholipases A2 and highlight the need for further study to analyze the complexity of antibody responses to these toxins. The study was however based on small animal sample (n = 2-3 per group), hence the interpretation of the findings is limited. The authors have attempted to acknowledge this limitation by stating that this was an exploratory study.

Additional comments

The small sample size and the test limited to 2 toxins from only one snake species should be acknowledged in the abstract.

---

## Round 0.3 · accepted · Accept

The authors have adequately addressed the reviewer comments and made the manuscript acceptable for publication.